# Yoga, an Appurtenant Method to Improve the Sports Performance of Elite Romanian Athletes

Rocsana Bucea-Manea-Țoniș [1], Dan Gheorghe Paun [2], Veronica Mindrescu [3,*] and Cristian Cătună [2]

[1] Doctoral School, National University of Physical Education and Sport, 060057 Bucharest, Romania
[2] Faculty of Physical Education and Sports, Spiru Haret University, 030045 Bucharest, Romania
[3] Faculty of Physical Education and Mountain Sports, University of Brasov, 500036 Brasov, Romania
* Correspondence: mindrescu.veronica@unitbv.ro

**Abstract:** Yoga is a very popular trendy sport all over the world. Since its establishment and promotion, yoga has mostly been practiced in social clubs and studios, often being performed during optional courses in colleges. Basic yoga instruction is generally absent in middle schools. This study investigated and assessed the viability of integrating yoga practice into the sports training program of elite Romanian athletes. The research methods used include documentation and data collection via an online survey in order to design a factor analysis with a structural equation model using SmartPLS software. The research assesses athletes' opinions about the benefits of yoga practice and its impact on post-traumatic stress disorder (PTSD). The results highlight the need to integrate yoga classes into educational institutions. The research has demonstrated that Romanian athletes use yoga in the pre- and post-competition phases to increase their focus, balance, muscle, and joint elasticity, create a positive attitude toward victory, manage emotional and post-traumatic stress disorder, visualize their performance in competition, and see themselves as winners. Overall, yoga is a successful strategy to enhance sports training and medical rehabilitation for stress disorders and post-traumatic diseases.

**Keywords:** yoga practice; sports training; technology; post-traumatic stress disorder; therapy

## 1. Introduction

Implementing social and physical isolation measures as a consequence of the spread of COVID-19 resulted in sudden and severe economic hardship, with marked decreases in both global trade and local small business activity [1]. The limitations implemented during the pandemic have impacted physical activity, which is associated with many health problems such as migraine, sleep problems [2], and respiratory, circulatory, and digestive problems. Therefore, it is important to evaluate one's level of physical activity and sleep in order to develop strategies for raising one's quality of life. Yoga is a therapeutic practice that can alleviate post-traumatic stress disorder such as that caused by our shared experience with COVID-19. We can take immediate steps to reduce the wave of disruption caused by the pandemic. Supporting overall mental wellness is crucial at this time. It seems natural to bring yoga back to help us cope with what is known as secondary traumatic stress, given that we are now experiencing collective trauma. Yoga-inspired methods can assist practitioners in overcoming COVID-19 and returning to their lives after the pandemic [3,4].

Several studies have demonstrated the additional advantages of yoga in stress reduction, chronic non-communicable disease prevention, and treatment. Technology, the Internet, and yoga exercises were the main means of psychological rectitude, social relations, academic training, and employment during the periods of quarantine and isolation caused by the COVID-19 pandemic [5]. Some authors have developed very easy-to-follow integrated yoga modules in the form of video recordings to be performed for disease prevention based on the scientific evidence that yoga improves respiratory and immunological functions [6,7]. Many athletes integrate yoga practice into training to increase their physiological skills, performance, and self-control ability and reduce post-traumatic stress disorder.

Therapeutic programs and therapies for stress reduction and the treatment of various stress- and lifestyle-related health disorders are all focused on mindfulness techniques. The cornerstone of the yoga path in achieving holistic health and well-being also includes ways and means of effective self-regulation. The consistent and correct practice of yoga postures as well as meditation can lead the elite athlete to superior self-awareness and self-regulatory skills compared to non-practitioners, which is reflected in higher levels of interoceptive awareness and decentering abilities [8,9]. The self-awareness, self-regulation, and self-transcendence (S-ART) framework in yoga is a solid pillar of training that elite athletes incorporate into their regular practice.

The mind–body practice in yoga classes combines physical exercise with a conscious inner focus on awareness of the self, breath, and energy [10]. Balance and oneness between body and mind can be achieved because yoga practice results in a physiological state that reverses the stress reaction. Yoga is used in sport to prevent injuries and improve performance by reaching peak physical fitness. Some studies have shown that a yoga intervention enhances athletes' flexibility, muscle strength, endurance, and cardiovascular performance [11,12], but also physiological health indicators such as heart rate, immune function, diastolic blood pressure, muscle discomfort [13], and mental fitness. Even if individuals only practice once a week, the benefits of yoga on flexibility can be felt within six weeks. Yoga training for just six weeks can lead to considerable increases in respiratory muscle strength and endurance in the yoga training group of young, healthy people. Practicing yoga for six weeks (45 min per day, five days per week) can improve cardiorespiratory endurance, abdominal muscle strength and endurance, and flexibility, and dramatically decrease body fat percentage in female students [14,15].

The development of physical stamina and flexibility as well as stress management, resilience, serenity, mind–body awareness, and spiritual/personal growth are important processes for the transformation of professional athletes. These processes have led to several improvements in the global health system's function, particularly in terms of physical and mental health and well-being [16].

Athletes enhance their performance by including yoga in their training programs. Performance can be achieved through attention, emotion, and elements of yoga along with cognitive, metacognitive, and procedural regulation strategies [17]. How the muscle gets accustomed to top-level training depends on many variables such as load, volume, frequency, mental–physical connection, contraction speed, work–rest ratio, and time spent on isometric exercise [18].

Taking into account all these aspects, the objectives of the present study were to identify and understand whether Romanian athletes and their trainers knew about the benefits of yoga practice. We also studied their views on associating elite training with yoga asanas, breathing, and mudras. At the same time, the research assessed whether they considered it important to use mantras and yoga meditation to reduce post-traumatic stress disorder. In addition, as we believe it is crucial to build more outreach initiatives and provide feasible, appropriate, and inexpensive opportunities to facilitate yoga inclusion in educational institutions, we asked for the opinions of athletes and teachers.

In order to achieve our goal, we designed an online survey for elite athletes and their trainers. The main research questions were:

Q1: Are athletes and their trainers aware of the benefits of yoga? Do these benefits significantly influence athletes' decision to integrate yoga practice into their training program?

Q2: Does yoga practice helps athletes cope with stress, anxiety, and post-traumatic stress disorder?

Q3: Is PTSD decrease an important factor that will influence the decision of trainers and athletes to include yoga in elite training?

Q4: Do the advantages of yoga affect the respondent's profile—their favorable attitude toward the inclusion of yoga classes in higher education institutions (such as medical schools and faculties of physical education and sports)?

Q5: Does the association of elite training and yoga practice improve athletic performance?

After data collection and selection, the authors implemented a structural equation modeling (SEM) factor analysis to assess the importance of each factor. Construct validation and content validation were both successful.

### 1.1. Benefits of Yoga and Elite Sports Training

The consistent practice of yoga, an aspect of traditional Indian culture and lifestyle, combined with a structured training program, helps practitioners track and increase their physical fitness levels. Yoga is beneficial for all athletes but is especially helpful in reducing injuries in sports that require quick movements, such as sprinting, tennis, basketball, and baseball. Yoga is thought to have therapeutic effects by enhancing vagal (parasympathetic) activation and decreasing the stress response of the sympathetic nervous system and the hypothalamic–pituitary–adrenal axis. Balance, flexibility, muscle strength, muscle endurance, and movement efficiency (coordination) are all key components of athletic performance that yoga closely resembles [14,15,19].

Yoga is believed to be connected with reduced aches and pains because its regular practice leads to the gradual relaxation of the muscles and connective tissues surrounding the bones and joints. Along these lines, the combination of the physical body, breath, and concentration while performing the postures and exercises unblocks the body's energy Nadis (channels) that open up, and the whole body's energy system reaches balance. Yoga breathing techniques focus on conscious prolongation of inhalation, breath retention, and exhalation. Pranayama reduces the amount of work required to breathe. Unlike shallow breathing, which only rehydrates the base of the lung, ventilation of the entire lung occurs [14,20]. This leads to an important increase in respiratory capacity, which will be reflected in higher performance achieved by athletes in training and competitions [21,22].

Yoga practice provides athletes with several tactics and procedures that will help them in both their academic and personal or social lives. Yoga enhances memory and cognitive functioning in target participants. This is because yoga can help athletes that live with high levels of stress. Practicing yoga over time will also be reflected in increased social-emotional competence [15].

As a result, both physical and mental health will be generally improved [23]. Yoga can be used in injury-prevention programs for a variety of reasons, including increased core stability, enhanced flexibility and range of motion, and improved relaxation. According to the findings of some studies, practicing yoga for four weeks significantly reduced body weight and increased leg and back strength in the experimental group in both intragroup and intergroup comparative tests. Godara et al. studied the effects of yoga training on 40 handball players aged 12–15 and found that the 10 weeks of yoga training considerably increased their back strength [14].

The study by Sojung et al. on 34 participants with an average age of 35 years examined the effects of eight months of yoga practice on their upper- and lower-body strength and found a substantial improvement in their leg strength. Calorie burning during physical activity as a combination of both isometric and isotonic exercises is one of the potential weight-reduction processes used in yoga therapy. Yoga helps decrease body fat percentage and increase fat-free mass after intervention [14,24]. Most long-term research using yoga interventions has found gradual and sustained decreases in body weight, body fat mass, body fat percentage, and BMI [25]. Adding yoga instruction to a structured training program significantly improved post-intervention flexibility and agility measurements, and dramatically improved balance following an intragroup intervention. This may be the result of correct training load optimization or prolonged yoga training.

Yoga brought a significant improvement in $VO_2$ max for adults who appeared to be healthy and who were assigned to a 12-week yoga training program due to better intercostal muscle control resulting from increased muscle endurance through yoga practice. The athletes' body composition and endurance measures were positively impacted [26].

### 1.2. Yoga Reduces Post-Traumatic Stress Disorder

Athletes are prone to injury at an epidemic rate. The likelihood of injury among athletes is considerably increased by the psychological and physical demands imposed on them by psychosocial stressors and training regimes [27,28].

Prolonged neurobiological recovery may increase the risk of recurrent musculoskeletal injury in sports-related concussions, which may have a longer neurobiological recovery time than clinical recovery time. Some researchers suggest a higher risk of musculoskeletal damage following a sports-related concussion. Compared to athletes without a history of sports-related concussions, players with a previous injury would experience a greater incidence of acute-noncontact injuries, but only female athletes would be affected by this relationship. Team sports coaches kept electronic records of athlete exposures and injury data throughout the competitive season, including injury characteristics: injury rates per 1000 exposures among athletes as well as incidence rate ratios (IRRs). Individuals who had previously experienced a sports-related concussion had an 87% higher risk of acute-noncontact lower-extremity injuries than participants who had not experienced such trauma. A past sports-related concussion did not affect the IRRs of acute-contact or overuse lower-extremity injuries. When compared to female athletes with no history of previous sports-related injury, young female players who had sustained a sports-related injury in the past 12 months were more likely to experience an acute-noncontact lower extremity injury during elite sports training [29].

The distinction between gender function and sports modality is evident in the recovery–stress state. It is recommended that female and male athletes in individual sports receive more specific attention. The recovery–stress balance in male athletes was not affected by their higher scores for the sports stress dimension compared to women [30].

Yoga can be successfully incorporated into players' athletic programs and, at the very least, can support the potential of a yoga intervention to reduce two important risk factors for injury: generalized fatigue and perceived susceptibility to injury. Examples of sports that benefit from yoga PTSD are football, handball, swimming, skiing, and baseball [27,28].

### 1.3. Yoga Improves Athletes' Performance

Yoga improves participants' sit-and-reach scores. According to the analysis made by Polsgrove et al., NCAA baseball and soccer players' flexibility and balance significantly improved after a 10-week yoga intervention compared to the control group. Examining the impact of yoga training on the flexibility of college athletes aged 18–24, the above authors concluded that, after 12 weeks of training, their flexibility was greatly improved. In addition, practicing yoga reduced lower back pain and increased muscle torque. An increase in ankle flexibility, knee extension, shoulder elevation, trunk extension, and trunk flexion was also noticed [14,24].

Ryan Giggs is an example of a footballer who reveals the importance of yoga techniques. At the beginning of his career, Giggs was tormented by injuries, which is why he had to find solutions to recover and shorten the recovery time; the solution was yoga, which allowed him to play until the age of 40 and count almost 1000 matches played for Manchester United. The six essential areas of training into which yoga is integrated are strength, speed/strength, flexibility/mobility, cardiovascular fitness/energy system development, recovery, and mental/emotional well-being [31].

Ernesto Cornejo believes that most injuries faced by football players are muscular. Cornejo was a professional footballer for teams such as FC Barcelona, SL Benfica, and Málaga CF. Using his own injury-related football experience, he began to use yoga techniques to relieve muscle tension. Noticing the benefits of this technique (the player feels more agile, lighter, and prevents numerous injuries), he retired at the age of 27 and created the Yogafit by EC center, and now he has football clients such as Borja Mayoral, Gonzalo Melero, Sergio Canales, Luis Alberto Romero, and Samu Castillejo [32].

Continuing the examples from football, we bring to attention the German National Football Team, which became World Champion in 2016 by incorporating yoga technique

training sessions with the help of yoga teacher Patrick Broome, who was introduced to the team by Oliver Bierhoff, a follower of yoga techniques. In addition to the physical benefits, we also add the mental ones, but the biggest gain was the fact that the players continued to apply these yoga techniques after the World Championship, being convinced of their support in sports activity [33].

Basketball legend LeBron James considers yoga an effective way to improve athletic performance. Basketball players need to be agile, flexible, and coordinated to prevent injury, and yoga can support performance at all levels [34].

### 1.4. The Necessity of Integrating Yoga into the Classroom

In schools, stress- and anxiety-related diseases are becoming a major public health concern. Most athletes have adopted yoga successfully. Some of them reported a range of psychological and physical advantages in addition to its general restorative effects, including greater mobility and flexibility, less pain or other physical afflictions, better posture, and better sleep. Some individuals have applied the new techniques, breathing or meditation, in their daily lives. This opportunity for self-responsible practice is a pleasant change. Several researchers have debated whether yoga should be a sport integrated into the educational system, whether it should be evaluated or graded, and whether it should be compulsory or optional. As young people who practice yoga can experience a variety of health benefits, including psychological and overall regenerative effects, the balance is inclined to introducing yoga classes because they can promote positive behavior by making individuals more aware of unhealthy tendencies [19,20].

The need to integrate yoga into the classroom so that students can practice it throughout their training is growing. Since yoga is an inclusive sport, it could be the responsibility and willingness of physical education instructors to actively participate and teach it. Including yoga in the educational system would have an important impact on the future of students. Yoga should be used in educational establishments, which would substantially encourage the acceptance of diversity, fostering an atmosphere of equality and inclusion in the learning environment in which they participate along with athletes. As a result, yoga is performed in a context of friendship, involvement, and impartiality, far from the actual climate manifested in schools, which, in many cases, results in conflicts, bullying, and other negative behaviors. Moreover, yoga improves the learning performance of students and is also considered a practice that reduces levels of forced competitiveness [21].

### 1.5. Technology That Could Facilitate the Inclusion of Yoga Practice in the Classroom

Although it has sparked numerous issues, technology (such as AI, blockchain, and IoT—Internet of Things) has had good effects on physical education and sports. It has been claimed that technology slows down sporting events or fails to meet the social needs of athletes. Consequently, the accuracy provided by technology proved to be quite helpful when making the right option and having solid evidence and arguments to support it. Monitoring a person's vital signs, the number of repetitions, and fitness level can ensure safety and proper training regimens. Exercise intensity is measured and adjusted in real time, the number of steps and repetitions are counted, the movement trajectory is examined, and athletes are provided with real-time feedback on the accuracy and correctness of their movements. These are some of the advantages of using technology in sports, and in yoga, as a particular case [30]. Blockchain and Internet of Things (IoT) technology make it possible to analyze athletes' performance in great detail. They also provide a solid foundation for research and innovation into novel theories, approaches, and techniques that could be successfully applied to sports. For instance, temperature, acceleration, sight-tracking sensors (such as BlazePod), and IoT-assisted energy harvesting devices for athletes are used to track their body temperature and exercise steps (IoT-EHDS). To offer a reliable Big-Data source and background for machine learning, this information is sent to the cloud under the privacy and anonymity provided by blockchain facilities. The information will be organized by ML algorithms, which will also be able to recognize and monitor

an athlete's fitness level or movement accuracy in real time. They all contribute to the evolution and advancements of the sports industry [35–37]. Through multimodal IoT, physical and physiological parameters related to sports injuries can be collected, analyzed, and assessed. The results can then be shared via blockchain with medical personnel so that they can manage and occasionally prevent sports hazards [38].

According to [39–43], developing a disruptive business model in physical education and sports (PES) will enable:

- Nonstop access to lectures/data for athletes all over the world based on blockchain, accessing information after payment. Some universities (such as Woolf University in Oxford and Cambridge) currently manage student–teacher collaboration using blockchain technology and smart contracts [43];
- Innovation in education: entrepreneurial universities strive to help athletes acquire transferrable skills for various sustainable activities;
- Management of transportation and accommodation facilities for athletes and staff, providing them with safe and convenient travel;
- Use of electronic payments for course participants;
- Digital marketing technique to promote sports.

Within the open-ended questions of our survey, athletes reported using online yoga and sports-specific lectures and videos. They also stated that implementing a blockchain disruptive business model in PES would be an appropriate and effective solution.

Overall, our literature review research emphasized many advantages of including yoga in the education and training of athletes. It also highlighted the advantages of using technology in managing sports training activities. Based on the knowledge gained, we decided to design a questionnaire to assess the state of the art in Romania.

## 2. Materials and Methods

### 2.1. Preliminary Phase of the Research

The research design was defined by the following topics: yoga, sports (Topic), 2019–2023 (Year of Publication), breath (All Fields), technology (All Fields), articles or review articles (Document Types), and all Web of Science Categories. We have studied many articles examining the benefits of yoga for human health and the association between yoga practice and sports. In Romania, yoga does not have a long history. Thus, we decided to analyze the level of awareness on this topic and its potential impact on elite athletes.

The study investigates how yoga can support sports performance as an associated method to provide the best possible sustainable system of education and training, having in mind the online teaching period during the COVID-19 pandemic.

The main research issues envisaged are:

- Identifying the reasons and benefits for which yoga is practiced in sports during the pre- and post-training phases.
- Determining the extent to which teachers, trainers, and athletes think that yoga can be used in elite sports training.
- Identifying the main ways in which yoga can reduce stress, anxiety, or post-traumatic stress disorder due to extreme competition, lack of rest, overtraining, and involvement in too many daily activities.
- Identifying the respondents' profiles and opinions about the introduction of yoga in specialized educational structures.

The main hypotheses of the research are:

**H1:** *The benefits of yoga practice will strongly influence athletes' attitudes toward yoga inclusion in their training program.*

**H2:** *Yoga practice helps athletes cope with stress, anxiety, or post-traumatic stress disorder both during competition and in daily life.*

**H3:** *Reducing post-traumatic stress disorder will positively influence the decision of teachers, trainers, and athletes to include yoga practice in elite sports training.*

**H4:** *The benefits of yoga influence the respondent's profile such as their positive attitude toward yoga inclusion in specialized educational structures (high schools, faculties of physical education and sports, medicine, etc.).*

**H5:** *Teachers, trainers, and athletes who are well-informed about the benefits of yoga think that incorporating yoga into an elite sports training program will improve athletic performance.*

*2.2. Design and Research Phase*

The research was conducted from the perspective of the survey applied to teachers and athletes from three universities, two in Bucharest (National University of Physical Education and Sport, and Spiru Haret University) and one in Brasov (University of Brasov), within the Faculty of Physical Education and Sport (PES). The research period was between 15 May 2022 and 15 October 2022. The population size was around 500 athletes and trainers from Bucharest and Brasov. We collected 250 responses, but 20 were invalidated as incomplete or with contradictory responses. Given that the minimum threshold for a representative sample is 218, we met this requirement. We cannot extrapolate the results to the entire population of Romanian athletes, but this preliminary study will allow and initiate new research with in-depth specific questions. Trainers, teachers, and athletes have responded to questions related to their profiles, and demographic and personal variables, but also to questions about the reasons for which yoga is practiced in sports (pre-/post-training), the methods of using yoga in elite sports, or whether yoga is effective in managing stress, anxiety, or post-traumatic stress disorder (PTSD) due to high levels of competition and overtraining in sports. The survey mostly consisted of closed-ended questions with multiple possible responses or specific options and was implemented online via Google Forms, obtaining the participants' consent to manage their responses in accordance with the GDPR framework. We also provided open-ended questions to initiate qualitative research.

The survey was inspired by other surveys such as those presented by different authors [44–46], but also by the practical experience of the main author who is a yoga instructor, and the practical experience of teachers and athletes who implemented sequences of different asanas in elite sports training.

The study variables were theoretically and practically specified at this point, and scales were selected to measure these variables, identify the information-gathering strategy, choose the data collection tool, and build the methodology to systematize information. The partial least squares (PLS) structural equation modeling method, which examines latent, formative, or reflective variables with simultaneous interactions even for smaller samples, was used to assess the data. Our model contains three latent formative constructs (Profile, PTSD, and Benefits of Yoga) and a reflective contract that assesses the opinions of teachers, trainers, and athletes about the introduction of yoga in elite training routines. These variables and their items are detailed in Table 1.

**Table 1.** Name, code, and significance of analyzed variables.

| Variables | Items | Description Significance |
|---|---|---|
| | Yoga practiced in the pre-/post-training sports phases has the following benefits: | |
| | Prevent9 | preventing injuries through physical and mental preparation for the competition |
| | Fitness9 | achieving performance by reaching the maximum level of physical fitness |
| | Mobility9 | achieving performance by increasing joint mobility and muscle elasticity |
| Benefits of Yoga | Endurance9 | achieving performance by increasing muscle strength and endurance |
| | Cardio9 | improving cardiovascular performance (heart rate, diastolic blood pressure) |
| | Imunity9 | improving the immune system |
| | Control9 | improving self-control and mental focus |
| | Pulmonary9 | increasing lung capacity and strengthening the diaphragm muscle |

Table 1. *Cont.*

| Variables | Items | Description Significance |
|---|---|---|
| | Balance9 | improving physical and mental balance |
| | BMI9 | weight control assessed by body mass index (BMI) |
| | How do you think yoga can be used in elite sports training routines? | |
| Yoga in Sports Performance | Energy10 | before matches (football, polo, volleyball, basketball, handball, etc.) or individual sports competitions, athletes can practice yoga breathing exercises (Breath of Fire) to energize their bodies |
| | Emotion10 | balance postures in yoga (practiced before training) help maintain mental/emotional balance during matches/competitions |
| | Performance10 | concentration postures in yoga (practiced before training) lead to superior performance in the specific training of each sport |
| | Acceptance10 | "ocean breath" (relaxation) exercises allow for easier acceptance of failure and remobilization for a new competition |
| | Visual10 | mental visualization (meditation) during relaxation in order to self-program the performance that can be achieved in the next competitions |
| | PTSD10 | reduces stress, anxiety, depression, or post-traumatic stress disorder (PTSD), facilitating the appropriate competitive mental state |
| | Yoga reduces stress, anxiety, or post-traumatic stress disorder | |
| PTSD | Competition11 | due to competition |
| | Unrest11 | lack of rest |
| | Daily11 | involvement in too many daily activities |
| Profile | Ytype | What types of yoga do you practice? |
| | Type14 | Do you believe that yoga practice is more effective? |
| | School16 | Do you consider it important to introduce yoga in specialized educational structures (high schools, faculties of physical education and sport, medicine, etc.)? |
| | Area17 | Work in the area: urban/rural |
| | Age20 | Specify your age |

## 3. Results

The analyzed data resulted in frequency, correlation, and graph tables, as well as a qualitative research regression model developed in the SmartPLS program.

### 3.1. Descriptive Statistics

The key findings are: most respondents practice Hatha (41.4%), asanas, pranas, mudras, and bandhas as part of their training routine (36.6%). Another 35% practice Ashtanga, Sivananda, Iyengar, and Kundalini Yoga. The rest practice Aerial, Vinyasa, and Kriya Yoga. Our respondents are gender-balanced: 57.6% are male and 42.4% are female. Most of them are young, under the age of 30 (52.4%). The number of respondents aged between 31 and 40 years (19.9%) is almost equal to the number of respondents aged between 41 and 50 years. Very few respondents are older than 50 years (6.3%). Most of the respondents are from urban areas (80.6%), and 19.4% are from rural areas. Many of them (50.8%) consider it important to introduce yoga in specialized educational structures (high schools, faculties of physical education and sports, medicine, etc.), 31.4% are neutral, and 17.8% do not consider that yoga would be an interesting subject to study in educational programs. Our respondents believe that yoga could be practiced by athletes in very short sessions during their moments of maximum detachment, with adapted postures (42.9%) in 2–3 sessions of 1 h each, followed by 30 min of density per week (18.3%); both variants could be practiced but require the judicious organization of time and personal activities (30.3%), and 8.4% do not consider that the practice of yoga is effective. Most of our respondents practice team sports (football, handball, basketball, volleyball, rugby, etc. 63.4%), followed by individual sports (athletics, gymnastics, dance, cycling, etc. 29.3%), winter sports (skiing, skating, sledding, hockey, etc. 14.1%), water sports (swimming, kayaking, water polo, surfing, diving, etc. 13.6%), racquet sports (tennis, badminton, table tennis, etc. 11.5%), mountain

sports (tourist orientation, climbing, skydiving, etc. 11%), defense sports (boxing, fencing, karate, etc. 12.6%), yoga (4.7%), and other sports (7.3%). During the pandemic period, most yoga classes were taught online through video recordings, video conferencing, and dedicated mobile apps. More than 50% of respondents continue to take yoga classes in the online environment or to get informed, but they state that the impact of asanas, and especially meditation, is much stronger when practiced outdoors or in clubs.

### 3.2. Path Analysis

SmartPLS is a reliable regression technique because it: (a) reduces the variance of endogenous construct residuals; (b) has only minor identification problems; (c) produces useful results even with small sample sizes; (d) primarily combines formative and reflective constructs [47]. PLS-SEM (partial least squares SEM) or path analysis should be used when the structural model is highly complex, the sample size is small, and the model incorporates both formative and reflective constructs [23,47]. PLS-SEM is the best method when the investigation focuses on theory development or prediction (with contributions to theory development). Predictive analysis and explanation of complex relationships are its main uses [48].

To ensure the validity of the survey, we check whether the questions contained contribute to constituting the significance of a statement/hypothesis from which it was started. When all items that make up a questionnaire correspond with each other's results, the questionnaire is reliable and consistent (overall score).

As the number of items (questions) increases, the value of Cronbach's alpha index also increases. However, it is pointless to preserve items whose contribution to the aggregate score is zero, minimal, or, on the contrary, goes in the opposite direction. One of the goals of item analysis is to identify these items, remove them, or modify them so that they can better reflect the measured feature.

It is cyclical in nature and operates by selecting things based on how they relate to the total score after assessing the relationships between items and between items and the overall score in turn. The value of Cronbach's alpha index, which can range between 0 and 1, serves as the primary criterion for this procedure. A scale should be as close to 1 as possible to be considered consistent, with 0.70 generally recognized as the upper limit. However, Cronbach's alpha cannot be less than 0.60.

### 3.2.1. Construct Reliability and Validity

In order to assess consistency through composite reliability, the study used SmartPLS [49] as indicated in Table 2. Composite reliability (>0.6), Cronbach's alpha, rho_A (>0.7), and average variance extracted (AVE > 0.5) are the authorized threshold values for a consistent model. According to Cronbach's alpha coefficients, the questionnaire items are acceptable for our study, which means that our hypotheses are grounded. The Benefits of Yoga variable shows extremely high values for all tests, including composite reliability (CR = 0.955 > 0.7), Cronbach's alpha (CA = 0.958 > 0.7), average variance extracted (AVE = 0.68 > 0.5), and rho_A (0.958 > 0.5). The Yoga in Sports Performance (YSP) variable shows extremely high values for all tests: CR = 0.958 > 0.7, CA = 0.958 > 0.7, AVE = 0.78 > 0.5, and rho_A = 0.960 > 0.5 (Table 2). The coefficient of determination or R-squared is 0.656, and, therefore, above the recommended minimum of 0.5. We can claim that the model explains 65.6% of the variation of the variable. BY, PTSD, and Profile variance explain 65.6% of YSP variance (Table 2 and Figure 1).

**Table 2.** Validation steps.

| Formative/Reflective Constructs | Composite Reliability (>0.7) | Cronbach's Alpha (>0.7) | AVE (>0.5) | rho_A (>0.5) | R-Squared (>0.5) |
|---|---|---|---|---|---|
| PTSD | - | - | - | 1 | 0.368 |
| Profile | - | - | - | 1 | 0.158 |
| Benefits of Yoga (BY) | 0.955 | 0.955 | 0.680 | 0.958 | |
| Yoga in Sports Performance (YSP) | 0.958 | 0.958 | 0.790 | 0.960 | 0.656 |

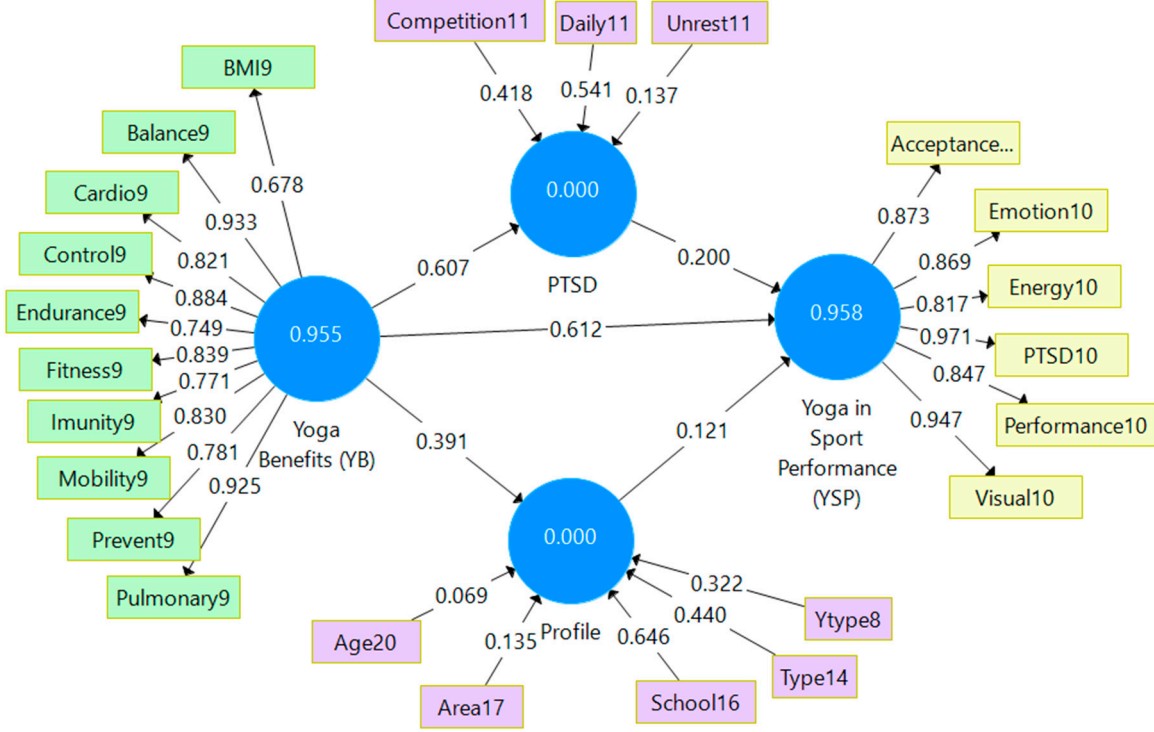

**Figure 1.** Cronbach's alpha analysis and path coefficients. Source: SmartPLS analysis (reprinted from a free version of SmartPLS software, version 3.3.9, created on 20 December 2022) [45].

### 3.2.2. Correlation between Variables

The latent variable correlation shows a very strong positive association between BY and YSP-0.781 (Table 3), and the path coefficient (0.612) also has a high value (Figure 1), meaning that H1 is accepted: the benefits of yoga practice will strongly influence athletes' attitudes toward yoga inclusion in their training program.

**Table 3.** Correlation coefficients.

| Latent Constructs | PTSD | Profile | BY | YSP |
|---|---|---|---|---|
| PTSD | 1 | | | |
| Profile | 0.392 | 1 | | |
| BY | 0.607 | 0.391 | 1 | |
| YSP | 0.619 | 0.439 | 0.781 | 1 |

Source: SmartPLS analysis (reprinted from a free version of SmartPLS software, version 3.3.9, created on 20 December 2022) [45].

Another medium positive correlation (Table 3) is observed between BY and PTSD (0.607) with the same value for the path coefficient, meaning that our second hypothesis (H2) is accepted: yoga practice helps athletes cope with stress, anxiety, or post-traumatic stress disorder both during competition and in daily life. A similar medium positive

correlation is noticed between PTSD and YSP (0.619) with a value of 0.200 for the path coefficient (Figure 1), meaning that our third hypothesis (H3) is accepted: reducing post-traumatic stress disorder will positively influence the decision of teachers, trainers, and athletes to include yoga practice in elite sports training.

A small positive correlation (Table 3) is observed between:

- BY and Profile (0.391) with the same value for the path coefficient (Figure 1), meaning that our fourth hypothesis (H4) is accepted: the benefits of yoga influence the respondent's profile such as their positive attitude toward yoga inclusion in specialized educational structures (high schools, faculties of physical education and sports, medicine, etc.).
- Profile and YSP (0.439) with a value of 0.121 for the path coefficient (Figure 1), meaning that our fifth hypothesis (H5) is accepted: teachers, trainers, and athletes who are well-informed about the benefits of yoga think that incorporating yoga into an elite sports training program will improve athletic performance.

Figure 1 shows that the loading factors for most of the BY variables have values greater than 0.7, which proves their impact on the BY weight. The very high values of the loading factor (LF) for YSP items, which are greater than 0.7, can also be noted. These high values ensure that the survey was well designed, and the items that form the two main variables (BY and YSP) are highly representative of our analysis [46]. It can also be seen that the loading factors for PTSD and Profile are smaller, explaining their small influence on the YSP variable and their low path coefficient.

### 3.2.3. Discriminant Validity

The calculation of discriminant validity was performed. It is defined as the extent to which a variable in the structural model empirically differs from other variables [49,50]. The model is statistically robust as the Fornell–Larcker and heterotrait–monotrait (HTMT) criteria are met. Most of the values obtained for Fornell–Larcker were less than 0.70. HTMT ratios should be <0.85 to achieve discriminant validity [51], and in our case, it is 0.778, meaning that all constructs were statistically differentiated from each other when taken two by two (Table 4).

**Table 4.** Discriminant validity.

| | | Fornell-Larcker | | | HTMT |
|---|---|---|---|---|---|
| Latent constructs | PTSD | Profile | BY | YSP | BY |
| Profile | 0.392 | | | | |
| BY | 0.607 | 0.391 | | | |
| YSP | 0.619 | 0.439 | 0.781 | - | 0.778 |

Source: SmartPLS analysis (reprinted from a free version of SmartPLS software, version 3.3.9, created on 27 January 2023) [46].

### 3.2.4. Model Fit

Model fit was assessed using approximate fit indices such as standardized root mean square residual (SRMR ≤ 0.05). In our case, SRMR has a value of 0.049, which is less than 0.05. Additionally, all SRMR, d_ULS, and d_G values of the estimated model are higher than the saturated model threshold (Table 5) [52].

**Table 5.** Model fit.

|  | Saturated Model | Estimated Model |
|---|---|---|
| SRMR | 0.046 | 0.049 |
| d_ULS | 0.633 | 0.728 |
| d_G | 0.740 | 0.744 |

Source: SmartPLS analysis (reprinted from a free version of SmartPLS software, version 3.3.9, created on 27 January 2023) [46].

### 3.2.5. Multicollinearity Analysis

The SmartPLS program calculated the variance inflation factor (VIF) of each construct to assess the significance of variables. The results are summarized in Table 6. There is no multicollinearity between variables as there are no VIF values higher than 5.

**Table 6.** Collinearity analysis.

| Variable | VIF | Variable | VIF | Variable | VIF |
|---|---|---|---|---|---|
| Acceptance10 | 4.991 | Daily11 | 3.027 | Prevent9 | 2.418 |
| Age20 | 1.04 | Emotion10 | 4.998 | PTSD10 | 4.998 |
| Area17 | 1.02 | Endurance9 | 3.084 | Pulmonary9 | 4.999 |
| Balance9 | 4.983 | Energy10 | 3.949 | School16 | 1.128 |
| BMI9 | 3.21 | Fitness9 | 3.067 | Type14 | 1.136 |
| Cardio9 | 4.416 | Imunity9 | 3.382 | Unrest11 | 3.647 |
| Competition11 | 2.326 | Mobility9 | 3.322 | Visual10 | 5 |
| Control9 | 4.414 | Performance10 | 4.932 | Ytype | 1.04 |

Source: SmartPLS analysis (reprinted from a free version of SmartPLS software, version 3.3.9, created on 20 December 2022) [46].

The *t*-test statistics are representative and the *p*-values for all 5 SEM regressions are less than the 0.05 threshold, showing again that our models were well designed (Table 7). Figure 2 provides an overview of the findings. The bootstrapping value of two-tailed *t*-tests was greater than 1.96.

**Table 7.** The *t*-test statistics and *p*-values of the bootstrapping analysis.

|  | Original Sample (O) | Sample Mean (M) | Standard Deviation (STDEV) | *t* Test Statistics (\|O/STDEV\|) | *p* Values |
|---|---|---|---|---|---|
| PTSD → YSP | 0.200 | 0.209 | 0.089 | 2.255 | 0.025 |
| Profile → YSP | 0.121 | 0.124 | 0.058 | 2.097 | 0.036 |
| BY → PTSD | 0.607 | 0.621 | 0.061 | 9.979 | 0.000 |
| BY → Profile | 0.391 | 0.416 | 0.063 | 6.200 | 0.000 |
| BY → YSP | 0.612 | 0.601 | 0.069 | 8.899 | 0.000 |

Source: SmartPLS analysis (reprinted from a free version of SmartPLS software, version 3.3.9, created on 28 January 2023) [45].

We can assume that the indicators of the constructs are significantly positively associated and that hypotheses H1–H5 are accepted after considering all the validation processes illustrated in Tables 2–7 and Figures 1 and 2. So, we can infer that yoga is a useful extra technique to provide a better training regimen for elite athletes, which is close to specialized training.

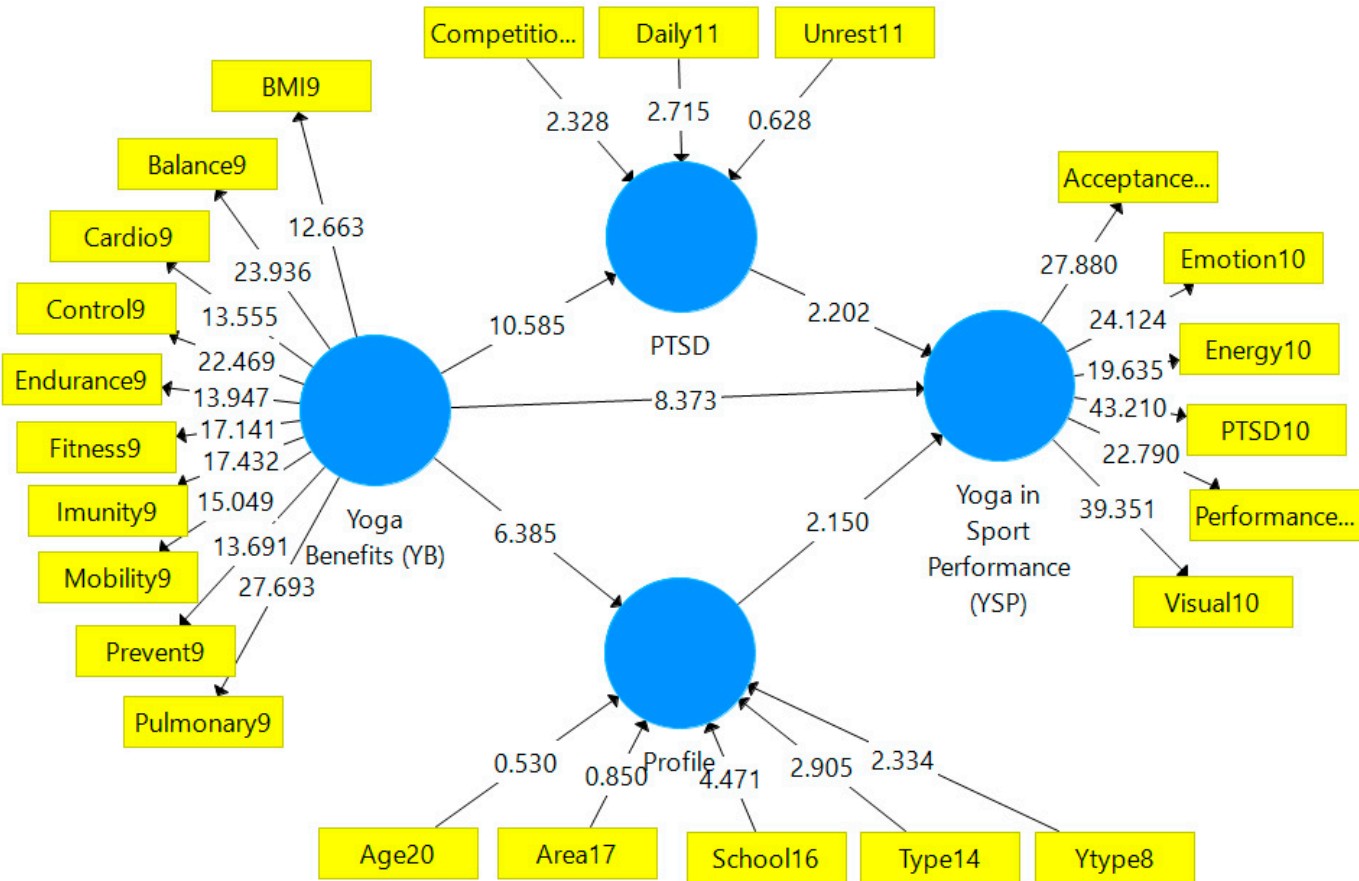

**Figure 2.** Bootstrapping. Source: SmartPLS analysis (reprinted from a free version of SmartPLS software, version 3.3.9, created on 29 January 2023) [46].

## 4. Discussion

Yoga has become a huge trend all over the world since it provides all practitioners with a lifestyle teaching technique combined with physical activity. Globally, there are around 30 million practitioners according to currently available information. Yoga is regarded as an injury-prevention technique that has many advantages, including the reduction of pain and tension and the calming and enlightening effect on the mind. The major documented benefits of yoga include an increase in mental and physical health [53,54], improved flexibility and body balance, reduced stress, an improvement in quality of life, weight management, and increased immunity [55].

Yoga is commonly used as a therapeutic method to promote both physical and mental health. This mind–body exercise can be a cost-effective and inclusive intervention used in schools to address mental health issues among young people. Research to date has primarily focused on neurotypical youth. However, there should be greater recognition of its effects on youth populations that are neurodiverse. The purpose of this research was to find out whether Romanian athletes knew about the positive impact of yoga when integrated into their daily practice. Yoga programs offered in schools can increase self-perception, subjective well-being, executive function, academic performance, and attention. The use of yoga in schools enhances mental health and cognition. It also paves the way for further studies and the creation of school-based yoga interventions [56].

Researchers have shown that yoga practitioners manifest more sustainable consumption than the general public. Sustainable consumer behaviors of yoga practitioners concerning sports marketing, business communication in sports, and sustainable marketing have not yet been studied. Future research on sports marketing and sustainable consumer behavior should take into account the major findings of the study, which are groundbreaking in many respects. Research findings show a beneficial relationship between yoga

practice and sustainable consumption. This situation can be explained by the fact that the level of awareness increases as the level of experience increases. Adopting sustainable consumption habits requires sustainability and awareness. People are willing to accept and use the ideas of sustainable development in their daily lives. Once someone performs a sustainable activity, they frequently repeat it. Therefore, sustainable conduct might last forever. Athletes who practice yoga have a high level of awareness because they apply its principles in their everyday lives. At this point, it can be argued that the principles and teachings of yoga are crucial instruments for fostering ethical consumer behavior.

With various reasons for participating, the three characteristics that most distinguished the yoga participant were spirituality, mind–body connection, and coping/stress management. Yoga-related studies and marketing materials need to consider how participants understand yoga and emphasize motivations that are appropriate for target subgroups to promote participation [57].

Our research has shown that yoga inclusion in specialized educational institutions (high schools, faculties of physical education and sports, medicine, etc.) is essential in the eyes of teachers, trainers, and athletes.

The following advantages of yoga practice during the pre- and post-training periods in sports are preparation of both body and mind for competition; injury prevention; being as physically fit as possible to achieve performance; increasing joint flexibility, muscle suppleness, as well as muscle strength and endurance to improve performance; increasing cardiovascular efficiency (heart rate, diastolic blood pressure); strengthening the immune system; enhancing retention and concentration; promoting physical and mental balance by strengthening the diaphragm muscle and lung capacity; measuring the body mass index (BMI) for weight loss. Yoga helps athletes who suffer from stress, anxiety, or PTSD due to overtraining, intense competition, lack of rest, or engagement in too many daily activities. Yoga develops mental toughness and detachment from pain and sorrow.

When athletes practice yoga "ocean breath" (relaxation) exercises before matches or individual sports competitions, they can reduce mental tension and energize their bodies. Before matches or individual sports competitions, athletes can practice yoga breathing exercises (Breath of Fire) to energize and nurture all the body's systems. After a game or competition, "ocean breath" (relaxation) exercises help the athlete regain homeostasis. They also make it easier to accept failure and help the athlete re-mobilize for the next match or competition. Other benefits of "ocean breath" (relaxation) exercises include: mental visualization (meditation) during relaxation helps the athlete self-program the performance that can be achieved in the next competition; setting good intentions during yoga Nidra (yogi sleep) can also be helpful to the athlete.

Balance postures in yoga (practiced before training) strengthen the triple extension chain that favors the correct execution of movements in team or individual sports; balance postures in yoga (practiced before training) help maintain mental/emotional balance during competitions; concentration postures in yoga (practiced before training) lead to superior performance in sports-specific training; asana practice improves core stability, which is crucial for injury prevention and athletic performance; yoga practice generally differs from other aversion therapies that emphasize the active engagement of only specific areas of the body because it requires the movement of numerous major and minor muscle groups at the same time and focuses on optimal muscular and skeletal alignment; an improved sense of balance and strength for whole-body movements is a typical result of consistent asana practice; since yoga increases joint range of motion and balance, it can improve performance in sports that require these qualities [14,24].

Although considerable advantages have been noted, it is challenging to identify the most beneficial yoga asana. Research on the impact of yoga on an athlete's range of motion should continue in the future. The dosage of yoga intervention may be quite high and may not be feasible or practicable for other people.

## 5. Conclusions

Yoga is a very popular trendy sport all over the world. Since its establishment and promotion, yoga has mostly been practiced in social groups and studios, often being performed during optional courses in colleges. After a comprehensive literature review, this study used an online survey to investigate and assess the viability of integrating yoga practice into the sports training program of elite Romanian athletes from three universities in Bucharest and Brasov. The findings have highlighted that athletes use yoga before and after competitions to improve their attention, endurance, balance, muscle elasticity, and joint flexibility. They also use yoga to manage their emotions and post-traumatic stress disorder, visualize their performance in competition, and see themselves as winners. Yoga practice can promote the development of a balanced outlook on life, a healthy physique, and a peaceful state of mind. Athletes incorporate yoga into their training to develop ingrained sports-related behaviors. Overall, yoga is a successful strategy to enhance sports training and medical rehabilitation for stress disorders and post-traumatic diseases. Technology provides sufficient support to keep track of an athlete's vital signs, fitness level, number of repetitions and sessions, level of strain, movement pattern, and activity in learning management systems.

## 6. Limitations

Our study is limited to three PES universities in Romania, but they are very important and representative of the whole country. However, we cannot extrapolate our results to the entire population of the country. This is a preliminary study.

Regarding technology, there are a lot of new methods and technologies used in PES but not used in this research, for example, gas leakage detection and pressure difference identification by the asymmetric differential pressure method; methods to evaluate and measure the power of pneumatic systems and their applications. They will be implemented in future research.

To understand how yoga helps people, a variety of yoga practices and nonspecific contextual aspects associated with yoga need to be taken into account. These exercises incorporate both contemporary and conventional ideas about yoga as a more comprehensive way of integrating yoga practice into elite sports training. These issues will be detailed as athletes become more proficient in yoga.

The absence of a long-term follow-up would be another drawback of this research. Studying the psychological effects of yoga on athletes due to meditation and developing a methodology to integrate yoga into elite sports training to improve their performance would be other challenging topics for future research. The dosage of yoga intervention may be quite high for some people and may not be feasible or practicable for other people. The novelty of this article is represented by the research regarding Romanian sport practicians' opinion on yoga inclusion in their training to reach higher performance and healthier behavior.

**Author Contributions:** Conceptualization, R.B.-M.-Ț., D.G.P., C.C. and V.M.; methodology, R.B.-M.-Ț. and V.M.; software, R.B.-M.-Ț. and C.C.; validation, R.B.-M.-Ț., D.G.P., C.C. and V.M.; formal analysis, R.B.-M.-Ț., D.G.P., C.C. and V.M.; investigation, R.B.-M.-Ț., D.G.P., C.C. and V.M.; resources, R.B.-M.-Ț., D.G.P., C.C. and V.M.; data curation, R.B.-M.-Ț. and D.G.P.; writing—original draft preparation, R.B.-M.-Ț., D.G.P., C.C. and V.M.; writing—review and editing, R.B.-M.-Ț., D.G.P., C.C. and V.M.; visualization, V.M. and C.C.; supervision, R.B.-M.-Ț. and D.G.P.; project administration, R.B.-M.-Ț., D.G.P. and V.M. All authors have read and agreed to the published version of the manuscript.

**Funding:** This research received no external funding.

**Institutional Review Board Statement:** Not applicable.

**Informed Consent Statement:** Informed consent was obtained from all subjects involved in the study.

**Data Availability Statement:** Not applicable.

**Conflicts of Interest:** The authors declare no conflict of interest.

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
