# Peer review of "Yoga, an Appurtenant Method to Improve the Sports Performance of Elite Romanian Athletes"

_sustainability, doi:10.3390/su15054264_

Round 1

Reviewer 1 Report

The authors claim that the purpose of their study was to research and assess the viability of yoga classes in high Romanian sports clubs. It is a very interesting study, and according to my opinion it is really important for feature research, since the practice of yoga is very important not only for physical health and performance, but also for whole-person wellness.

I think however, that the structure of the paper is not really helpful for the reader to conceive the purpose of the study. Actually, it is like two different papers, a research paper and a review are linked in a large paper. There is two much information that is difficult to follow. After reading the abstract, I was not sure if it is a research paper, since there are mentions about findings, but there is no clear methodology described and the most important, there are no numbers indicating statistical analysis. In the results section however, I found out that the analysis used is path analysis! it is a pity not to mention such a high class analysis in the abstract.

It was also difficult to find the purpose of the study, since the introduction section is rather chaotic. As researchers, looking for information from a study, we have set our minds to a certain structure of a paper. That is, the reader is seeking for the purpose of the study about the end of the introduction section. The present paper however has not just an introduction, but also another section called "theoretical framework"?! Why this theoretical framework is not a part of the introduction? In a similar way, why the research hypothesis are listed in the Methodology and not in the introduction section?

In my opinion, the paper will be much improved, if the authors create a strict structure corresponding to a research paper and if they rewrite the introduction in a shorter version. 

Author Response

I have attached the document with the requested changes

Reviewer 2 Report

Dear Authors

You have written an interesting paper which researched and assessed the viability of yoga classes in high Romanian sports clubs.

2nd paragraph of the introduction  - several statements regarding the covid need to be backed up by references. Especially when you state "numerous studies" and there is no reference!

3rd paragraph / the whole paragraph with just 2 references at the end. Please add references after every strong/relevant statement! And this goes for the whole paper. 

Line 88 / delete .

The theoretical framework is well described. However, post-traumatic stress disorder is barely mentioned. Please add additional info and create an additional subheading for it in the theoretical framework. 

Line 300 - how does this item fit into the rest of the content/bullet points? elaborate

In research issues and hypotheses, there is no mention of technology that you focus so much on. Elaborate 

Design / report how did you determine your sample size?

How long was the survey available / report dates / 

Please describe the statistics section better. How did you check for the normality of data? Report exact tests and posthoc tests used. Report significance values. 

The limitations of the study paragraph is missing at the end of the discussion.

Conclusions are too general and you can not generalise your findings from a sample of 250 surveys to the whole country. Therefore, acknowledge this in the limitations and amend the conclusion. 

The references list needs to be updated and done in the correct reference style. Amend

Overall an interesting study. However, some more work needs to be done by the authors. 

Kind regards 

Author Response

(The authors gave the same response as above.)

Reviewer 3 Report

This paper presents a test of viability of yoga classes in high Romanian sports clubs, using a range of research methodologies, including the documentary method, collecting data using an online survey and analyzing it with SmatPls software. The technology provides adequate assistance for tracking a person's vital signs, their degree of fitness and repetitions, their level of exertion, their  movement trajectory and their activity on educational platforms. This paper has several limitations and the standard is not enough, and address the following items would result in a good paper,

1.  The literature review is not thorough about the application and the contributions. To highlight the contributions, it suggests reorganizing the section of the related work with real applications. It is recommended to read more related works and consider discussing their application scenarios in the introduction and discussion, for example, you can read these following articles:  Gas Leakage Detection and Pressure Difference Identification by Asymmetric Differential Pressure Method; Methods to Evaluate and Measure Power of Pneumatic System and Their Applications.

2.  The contribution of this paper is not clear. It suggests revising the contributions section and making these points clear and strong.

3.  The quality of the Figures should be improved and readable for the readers.

4.  Maybe it is better to discuss the possibility to improve the scope using online estimation in the introduction, for example, Gas Flow Measurement Method with Temperature Compensation for a Quasi-Isothermal Cavity.

5.  It is recommended to present in the first section so that it can highlight the specific scope of this article. The meaning of the assessment experiment should be highlighted.

6.  There should be a further discussion about the limitation of the current works, in particular, what could be the challenge for its related applications. To let readers better understand future work, please give specific research directions.

Author Response

(The authors gave the same response as above.)

Reviewer 4 Report

The manuscript address an important subject, however these are some concerns

·        It is unclear whether this is a research article or a review

·        The structure is similar to a review article, for example, the introduction is divided into different sections also the methods

·        The objective and the write up is unclear (it needs to be written in academic writing style)

·        The method section is very long and unclear, the data analysis is unclear

Author Response

(The authors gave the same response as above.)

Round 2

Reviewer 1 Report

There are substantial improvements in the document and I believe that the manuscript should merit publication.

I feel as a redundancy to write both research questions and research hypothesis in different parts of the manuscript. I see however, that the authors are using the research hypothesis in the description of their results.

Author Response

Thank you very much for the evaluation support and very useful recommendations. They really helped us to improve the article.

Reviewer 2 Report

Dear Authors,

Thank you for addressing my questions and suggestions fully. The corrections make your paper clearer. 

Regarding the question/guidance: I would leave this section in as it is a part of your data/questionnaire.

Overall, I recommend acceptance.

Kind regards

Author Response

(The authors gave the same response as above.)

Reviewer 4 Report

The manuscript was improved

Author Response

(The authors gave the same response as above.)
